# Farmers' Awareness of the Low Yield of Conventional Rice Production in Ayeyarwady Region, Myanmar: A Case Study of Myaungmya District

**Soe Paing Oo [1,2]**

[1] Graduate School of International Development (GSID), Nagoya University, Nagoya 464-8601, Japan; soe.paing.oo@b.mbox.nagoya-u.ac.jp or soepaingo@gmail.com; Tel.: +81-90-42452575

[2] Department of Agronomy, Yezin Agricultural University (YAU), Nay Pyi Taw 15013, Myanmar

**Abstract:** The Ministry of Agriculture and Irrigation introduced the Good Agricultural Practices (GAPs) of rice in 2008. The adoption rate of GAPs is still low. As the first step of the adoption process, this study investigates farmers' awareness of the low yield of conventional rice production. Based on the data of 315 farmers collected from a field survey conducted from July to August 2018 in Myaungmya District, Myanmar, and by applying the cluster analysis and binary logit model, the study found that farmers' awareness was low for the aspects of farmer management and Ministry management. The finding of most interest is that farmers with more experience, higher income level, larger farmland size, and receiving agricultural information were associated with low awareness. Farmers with more farming experience were satisfied with the return of rice from conventional production. Some farmers received a higher total income from crop production because of a larger farmland size, and they are less aware of the low yield of conventional rice production. Even though farmers received agricultural information, they could not apply the information to rice production. Farmers' awareness of the low yield can be increased through developing extension services programs to distribute useful information on rice production effectively.

**Keywords:** awareness of low yield; conventional rice production; good agricultural practices (GAPs); Myanmar

## 1. Introduction

In Myanmar, rice is a major food crop that occupies about 61% of the total sown area. Myanmar currently has multifaceted problems in agriculture, particularly the low productivity of rice, which has diminished the economy of the country. The national average yield of rice was less than 4 tons per hectare from 2001 to 2006. The Ministry of Agriculture and Irrigation (MOAI) decided to promote the growth of the rural economy through sustainable development of the rice sector. MOAI set up 5 tons per hectare as the national target yield of rice in 2007 [1].

MOAI introduced the application of Good Agricultural Practices (GAPs) of rice as an agricultural policy in 2008 to reach the target yield (5 ton/ha) [1]. The Food and Agriculture Organization (FAO) of United Nations originally created GAPs in 2008, and it has been implemented in many countries [2]. GAPs is a set of practices that address environmental, economic, and social sustainability for on-farm processes and results in safe and high-quality food and nonfood products [3]. GAPs of rice is a package of improved technologies, and it boosts the yield of rice [4]. In Myanmar, MOAI composed GAPs of rice with 14 components, from seed selection to harvesting [1]. Despite MOAI efforts, the adoption rate of GAPs of rice in 2016 was low (16.57%), and the national average yield of rice in 2016 decreased from 3.97 in 2015 to 3.77 tons per hectare [5]. In other words, most of the farmers cultivate rice by conventional methods [6].

It is necessary to understand the technology adoption process of farmers so that the GAPs of rice can disseminate in Myanmar. There are five steps of adoption: awareness, interest, trial, evaluation, and adoption [7]. Among them, awareness is an essential factor for the dissemination of environmental knowledge and communication of its fundamental elements [8]. Therefore, there is a need to study farmers' awareness before attempting the adoption of new technology [7]. As the first step of the adoption process, farmers must be aware of problems that relate to farming practices. If farmers notice their problems, they will search for the proper solution to the problems. Most of the approaches to study technology adoption by farmers do not investigate actual problems (farmers' awareness of the low yield) before the technology is introduced to farmers, such as [9–16].

A better understanding of the features of farmers' awareness and the factors influencing the awareness of farmers is needed to formulate appropriate agricultural policies and programs to cope with a low yield of conventional rice production. Therefore, the objectives of the present study are (i) to clarify the features of farmers' awareness of the low yield of conventional rice production and (ii) to analyze determinants of farmers' awareness of the low yield of conventional rice production.

## 2. Materials and Methods

### 2.1. Study Area and Sampling

Even though rice can grow in any state and region of Myanmar, the Ayeyarwady Region is famous for rice production because of its suitable fertile soil and favorable weather conditions for rice cultivation. This region tops the sown area of rice (28.29% of the total area) among 14 states and regions [6]. Two-thirds of the entire arable land in this region is under rice cultivation. This region is, therefore, known as the rice bowl or granary of Myanmar. MOAI introduced GAPs of rice in this region in 2008 and has encouraged farmers to use this technology [6]. This region has favorable soil and environmental conditions to apply the GAPs of rice in both wet and dry seasons (the wet season lasts around five months from July to November, while the dry season last approximately four months from December to March) of rice production [6].

The Ayeyarwady Region consists of six districts: Myaungmya, Pathein, Hinthada, Maubin, Pyapon, and Labutta. In the first stage, Myaungmya District was purposely selected based on two criteria. One was the average yield of rice in 2016. The average yield of Myaungmya District was 3.21 tons per hectare, and it was nearly similar to the yield of the Ayeyarwady Region (3.46 ton/ha). The other is the number of GAPs trainings for farmers in 2016. The number of GAPs trainings in this district was 12, which is lower than that of this region (14.17) [17]. In Myaungmya District, there are three townships: Myaungmya, Einme, and Warkhema (Figure 1). In the second stage, this study randomly selected three villages from each township, and then 35 farmers from each village were selected by landholding size. As a result, the total sample for this study was 315 farmers (Table 1).

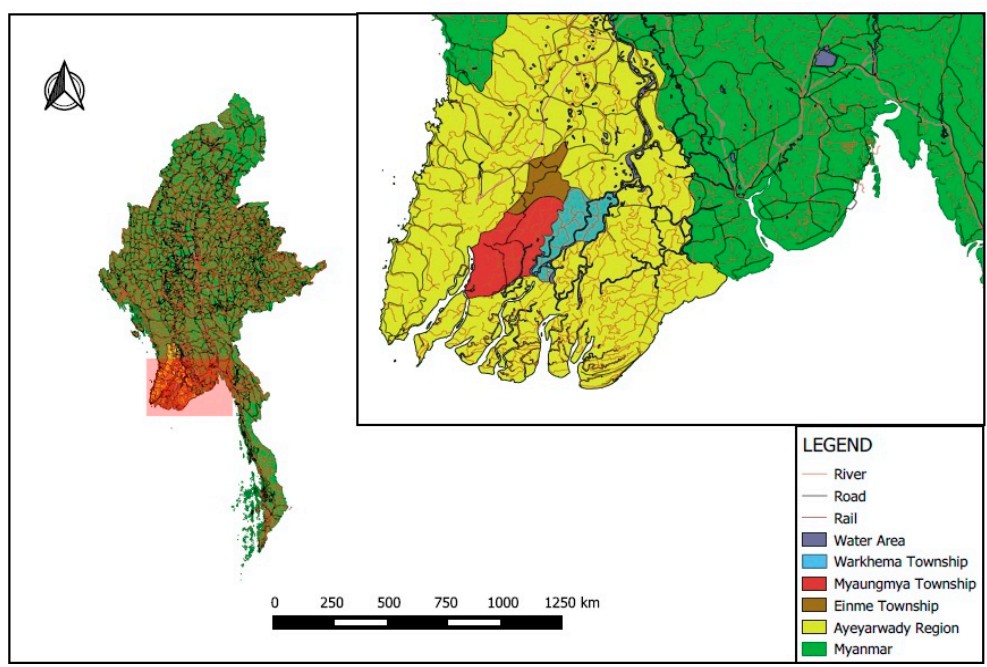

**Figure 1.** Location of the study area. Source: Author.

**Table 1.** List of sample villages and the number of respondents.

| Township (Number of Village Tracts *) | Name of Sample Village (Number of Households Who Cultivate Rice) | Total Respondents by Landholding Size ** | |
|---|---|---|---|
| Myaungmya (98) | Ma Dawt Pin (163) | 105 | 34 (S) |
| | Kyon War (242) | | 29 (M) |
| | Tha Pyay Chaung (215) | | 42 (L) |
| Einme (97) | Hpa Yar Gyi Kone (479) | 105 | 39 (S) |
| | Ye Thoe (275) | | 41 (M) |
| | Gone Hnyin Tan (242) | | 25 (L) |
| Warkhema (125) | Thea Kone (253) | 105 | 75 (S) |
| | Kyar Hpyu (382) | | 21 (M) |
| | Au Kyun Taw Gyi (345) | | 9 (L) |
| Total | (2596) | 315 | |

Source: [17]. Note: (1) * 5–12 villages organize village tract, and (2) ** S = small farmers who own less than 5 acres of farmland, M = medium farmers who own 5–10 acres of farmland, and L = large farmers who own >10 acres of farmland.

## 2.2. Framework and Variables

In the present study, farmers' awareness of the low yield of conventional rice production is predicted to be influenced by several factors. These include personal characteristics, farming characteristics, economic characteristics, institutional characteristics, and location (Figure 2).

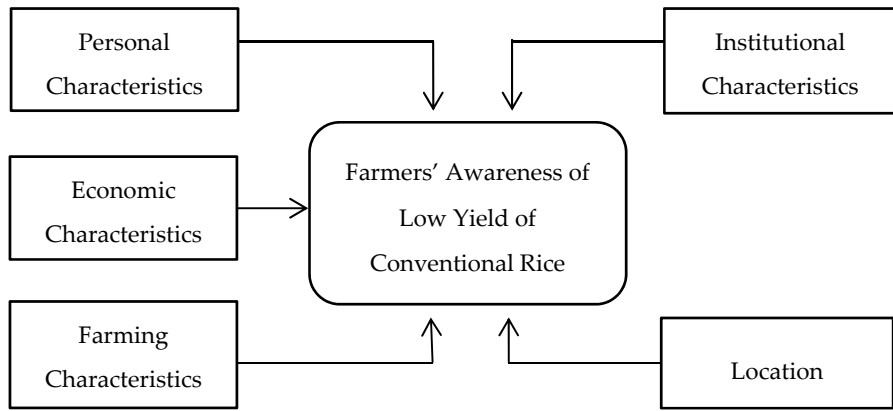

**Figure 2.** Conceptual framework of the study.

In this study, age, gender, marital status, education, farming experiences, and household size represent personal characteristics. Economic characteristics consist of access to credit and income from crop production. There are two variables of farming characteristics: farmland size and active labor force. Contact with extension workers, receiving agricultural information, and membership in local farmers' associations represent institutional characteristics. The location consists of three townships: Myaungmya, Einme, and Warkhema (Table 2).

In the study, the selected variables are based on previous studies. Education, farmland size, and annual income were positively correlated with awareness of the environment, while age showed a negative correlation [18]. Furthermore, age, gender, education, and farm size were determinants of farmers' awareness of ecosystem services [19]. However, marital status did not correlate with the awareness of ecosystem services. Family size was negatively associated with farmers' awareness of environmental degradation [20]. Age, education, and membership of community organizations significantly influenced awareness of land, soil, and water conservation practices [21]. Nevertheless, household size, family labor, access to agricultural advice, and access to agricultural credit were nonsignificant variables in their study. Age and membership in local farmers' associations were determinants of awareness and adoption of improved rice varieties [22]. Characteristics such as experiences of growing pigeon pea and contact with government extension agents were selected to analyze technology awareness and adoption in the case of pigeon pea varieties in Kenya [16]. However, the yield of rice ultimately varies from one location to another because of unexpected conditions. Therefore, farmers' awareness of low yield in one place will differ from that of another. Climate change of one region differs from that of another, and so does awareness [8].

**Table 2.** Description of independent variables.

| Independent Variables | Description | Symbol |
|---|---|---|
| **Personal characteristics** | | |
| Age | Age of household head (years) | *AGE* |
| Gender | 1 for male; 0 otherwise | *GEN* |
| Marital Status | 1 for married; 0 otherwise | *MST* |
| Education | Years of formal schooling | *EDU* |
| Farming experiences | Years of experience in farming | *FEXP* |
| Household size | Number of household members | *HHSIZE* |
| **Economic characteristics** | | |
| Access to credit | 1 for the household head has access to credit; 0 otherwise | *CRE* |
| Income from crop production | Level of annual income from crop production: 1 for low (<6,000,000 kyats), 2 for medium (6,000,000 to 10,000,000 kyats), 3 for high (>10,000,000 kyats) | *INC* |
| **Farming characteristics** | | |
| Farmland size | Size of farmland owned by household in acres | *FSIZE* |
| Active labor force | Number of household members who are actively involved in rice production | *LAB* |

**Table 2.** *Cont.*

| Independent Variables | Description | Symbol |
|---|---|---|
| **Institutional characteristics** | | |
| Contact with extension workers | Number of meetings per year (2017) | *EXT* |
| Receiving agricultural information | 1 for received; 0 otherwise | *INF* |
| Membership in local farmers' association | 1 for member; 0 otherwise | *MEM* |
| **Location** | | |
| Einme township | 1 for the farmer who lives in Einme township; 0 otherwise | *LOCE* |
| Warkhema township | 1 for the farmer who lives in Warkhema township; 0 otherwise (Myaungmya township as a base case) | *LOCW* |

### 2.3. Data Collection

Awareness is defined as a state of knowing and being informed of something [23]. Awareness is the state or ability to perceive, to feel, or to be conscious of events, objects, or sensory patterns [24]. Generally, awareness is the state or quality of being knowledgeable about something. Awareness refers to the concern for environmental problems [25]. When one realizes an environmental issue, awareness will improve and increase understanding. In the present study, awareness is defined as a state of knowing the reasons for the low yield of conventional rice production. If farmers properly know the reasons for the low yield of conventional rice production, they can go through the process of adopting appropriate technologies using the GAP of rice.

Based on a pilot survey and key informant interviews on the causality of low yield, the questionnaire for farmers' awareness consisted of ten statements, covering three aspects of awareness: general risks, farmer management, and Ministry management (Table 3). Awareness of general risks is to measure farmers' awareness of risks in farming. This aspect of awareness is essential because when farmers are aware of the risks faced by them, they are likely to change their farming practices. The awareness aspect of farmer management is to measure farmers' awareness of their management of farming practices. If farmers are aware of difficulties in rice production, they will find out the appropriate solution to improve their income from rice production. The aspect of Ministry management is essential because if farmers are aware that there are problems with the Ministry's management, the Ministry may be pressured to provide suitable policies, services, and agricultural inputs for farmers.

According to previous studies, awareness can be measured by two approaches. The first one is using Likert scale measures. This approach has already been used by previous studies such as in a climate change study [26], adaptation study [27], awareness of technology studies [10,28], and an ecosystem services study [13]. The second approach is using dichotomous choices ("yes" or "no") such as is found in [8] and [15]. This study followed the first approach (using the Likert scale) because this approach is more appropriate for obtaining a detailed answer from the respondent that furthermore includes a neutral response option.

In this study, farmers' awareness was measured directly by using the Likert scale (Likert scale is a type of rating scale to measure farmers' awareness; with this scale, respondents are asked to rate the statements according to their level of awareness) ranging from 1 to 5 (1 = strongly disagree, 2 = disagree, 3 = neither agree nor disagree, 4 = agree, 5 = strongly agree). Though five levels measure farmers' awareness, the Likert scales were converted to two categories: if the Likert score was less than 4, it was categorized as "not aware of the statement", while if the value was equal to or greater than 4, it was categorized as "aware of the statement". Cronbach's alpha (Cronbach's alpha is a measure of internal consistency; that is, how close a set of items is related as a group) was used to examine the strength of the interior flexibility and reliability of the awareness statements.

**Table 3.** Aspects and statements to measure farmers' awareness.

| Aspect | | Statement |
|---|---|---|
| General risks (natural condition, price, and human) | AW1 | Climate change (heavy rain and flooding) affects yield loss. |
| | AW2 | Less attention is paid to rice production due to the small profit. |
| | AW3 | Knowledge of rice production technology is inadequate. |
| Farmer management | AW4 | It is challenging to hire the required number of laborers when necessary. |
| | AW5 | Farmers cannot plant and harvest rice at the right time. |
| | AW6 | Soil fertility is becoming more inadequate for cropping. |
| | AW7 | Farmers do not use the adequate and correct amount of farmyard manure (FYM) and fertilizers. |
| Ministry management | AW8 | Agricultural policies of the Ministry of Agriculture and Irrigation are unstable. |
| | AW9 | Agricultural extension services are not helpful for farmers. |
| | AW10 | Quality seed is not sufficiently available for farmers. |

Note: AW = awareness.

## 2.4. Data Analysis

Descriptive analyses such as average, standard deviation, and percentage were used for categorizing and describing the variables whenever possible. The analysis of variance is applied to compare farmers' awareness among three townships. Moreover, the cluster analysis technique was used on survey data to define groups of farmers under similar awareness. It was also applied to investigate the proper model used in the analysis.

Since the dependent variable was "aware" or "not aware", the binary logit model was used for Objective (2). The binary logit model was also used to analyze the determinants of farmers' awareness [8,9]. The response of the farmer to "aware" or "not aware" of the low yield of conventional rice production can be written as follows:

$$Y_i = \begin{cases} 1 \ (aware) \\ 0 \ (not \ aware) \end{cases} \tag{1}$$

Suppose $P_i$ = probability of being aware and $1 - P_i$ = probability of not being aware.

$$P_i = \frac{1}{1 + e^{-Zi}} \tag{2}$$

$$1 - P_i = \frac{1}{1 + e^{Zi}}, \tag{3}$$

The equation for the binary logit model could be used as follows [29]:

$$\frac{P_i}{1 - P_i} = e^{\beta_i X_i + u_i} \tag{4}$$

$$ln\left(\frac{P_i}{1 - P_i}\right) = \beta_i X_i + u_i, \ i = 1, \ 2, \ 3, \ \ldots, \ n \tag{5}$$

where $X_i$ is the set of independent variables, $\beta_i$ is coefficient of independent variables, and $u_i$ is error term.

The study employed the following binary logit equation to determine the factors affecting farmers' awareness of the low yield of conventional rice production:

$$ln\left(\frac{P_i}{1-P_i}\right) = \begin{aligned} &\beta_0 + \beta_1 AGE + \beta_2 \ GEN + \beta_3 \ MST + \beta_4 \ EDU + \beta_5 \ FEXP + \beta_6 HHSIZE + \beta_7 \ CRE \\ &+ \beta_8 \ INC + \beta_9 \ FSIZE + \beta_{10} \ LAB + \beta_{11} \ EXT + \beta_{12} \ INF + \beta_{13} \ MEM + \beta_{14} \ LOCE \\ &+ \beta_{15} \ LOCW + u_i \end{aligned} \tag{6}$$

## 3. Results and Discussion

### 3.1. Characteristics of Farmers

A summary of farmer characteristics is shown in Table 4. The average age of household heads was 50.25 years. Most of the farmers were male, and they were married. Their average education level was 5.57 years. On average, farmers managed 9.69 acres of farmland and experienced farming for 25.56 years.

The average household size was 4.51 persons, and the active labor force per household was 3.39 persons. The average annual income from crop production was 8,004,010 kyats (kyat is the currency of Myanmar; 1 kyat = 0.00067 USD (1 USD = 1530 kyats, as of 9 December 2019). Farmers received agricultural information and could access credit for rice production. The average number of contacts with extension workers was 2.87 times per year. Half of the farmers were members of local farmers' associations.

### 3.2. Farmers' Awareness of Low Yield of Conventional Rice Production

Farmers' awareness of low yield of conventional rice production is shown in Table 5. Since Cronbach's alpha for the measurement of awareness was 0.75, the data on farmers' awareness scores were reliable for the analysis. The majority of farmers were aware of some reasons for the low yield of rice production. However, such awareness varied from statement to statement. According to the results of t-tests, the mean score was significantly different in each statement between "aware" and "not aware".

**Table 4.** Descriptive summary of farmers' characteristics.

| Farmers' Characteristics | Number of Farmers = 315 | |
| --- | --- | --- |
| | Average | Std. Dev. |
| Age (year) | 50.25 | 12.576 |
| Gender (% of male) | 97.46 | 15.8 |
| Marital status (% of married) | 95.24 | 21.3 |
| Education (year) | 5.57 | 3.309 |
| Farming experiences (year) | 25.56 | 13.706 |
| Household size (person) | 4.51 | 1.607 |
| Access to credit (%) | 91.74 | 27.6 |
| Income from crop production (kyat/year) | 8,004,010 | 11,244,539 |
| Farmland size (acre) | 9.69 | 13.386 |
| Active labor force (person) | 3.39 | 1.427 |
| Contact with extension workers (number) | 2.87 | 3.658 |
| Receiving agricultural information (%) | 87.94 | 32.6 |
| Membership in local farmers' association (%) | 45.71 | 49.9 |
| Location; Einme township (%) | 33.33 | 2.7 |
| Location; Warkhema township (%) | 33.33 | 2.7 |

Source: Field Survey Data (2018). Note: Std. Dev. = standard deviation.

**Table 5.** Comparison of mean values of farmers who were not aware and aware in each statement of awareness.

| Aspect | Statement | Not Aware (<4) | | | | Aware (≥4) | | | | *t*-Value |
|---|---|---|---|---|---|---|---|---|---|---|
| | | Respondents | | Likert Scale | | Respondents | | Likert Scale | | |
| | | Number | Percentage | Mean | Std. Dev. | Number | Percentage | Mean | Std. Dev. | |
| General risks | AW1 | 25 | 7.9 | 2.24 | 0.78 | 290 | 92.1 | 4.68 | 0.52 | 0.001 *** |
| | AW2 | 75 | 23.8 | 2.41 | 0.79 | 240 | 76.2 | 4.51 | 0.51 | 0.001 *** |
| | AW3 | 27 | 8.6 | 2.59 | 1.05 | 288 | 91.4 | 4.68 | 0.52 | 0.001 *** |
| Farmer management | AW4 | 105 | 33.3 | 1.95 | 0.73 | 210 | 66.7 | 4.50 | 0.63 | 0.001 *** |
| | AW5 | 65 | 20.6 | 1.85 | 0.75 | 250 | 79.4 | 4.55 | 0.52 | 0.001 *** |
| | AW6 | 57 | 18.1 | 1.98 | 0.88 | 258 | 81.9 | 4.60 | 0.56 | 0.001 *** |
| | AW7 | 63 | 20.0 | 2.14 | 0.76 | 252 | 80.0 | 4.58 | 0.50 | 0.001 *** |
| Ministry management | AW8 | 137 | 43.5 | 2.19 | 0.80 | 178 | 56.5 | 4.47 | 0.54 | 0.001 *** |
| | AW9 | 38 | 12.1 | 2.39 | 0.86 | 277 | 87.9 | 4.65 | 0.52 | 0.001 *** |
| | AW10 | 53 | 16.8 | 2.55 | 0.82 | 262 | 83.2 | 4.65 | 0.51 | 0.001 *** |

Source: Field Survey Data (2018). Note: (1) AW1 = Climate change (heavy rain and flooding) affects yield loss. (2) AW2 = Less attention is paid to rice production due to the small profit. (3) AW3 = Knowledge of rice production technology is inadequate. (4) AW4 = It is challenging to hire the required number of laborers when necessary. (5) AW5 = Farmers cannot plant and harvest rice at the right time. (6) AW6 = Soil fertility is becoming more inadequate for cropping. (7) AW7 = Farmers do not use the adequate and correct amount of FYM and fertilizers. (8) AW8 = Agricultural policies of the Ministry of Agriculture and Irrigation are unstable. (9) AW9 = Agricultural extension services are not helpful for farmers. (10) AW10 = Quality seed is not sufficiently available for farmers. (11) Std. Dev. = standard deviation. *** = significant at 1% level.

### 3.2.1. General Risks (AW1 through AW3)

- Among the three statements concerned, in terms of percentage, with farmers' awareness of AW2 (Less attention is paid to rice production due to the low profit), awareness was relatively low (76.2% of farmers).
- Farmers were more aware of AW1 (Climate change) and AW3 (Knowledge of rice production technology is inadequate).

### 3.2.2. Farmer Management (AW4 through AW7)

- Among the 4 statements concerned, in terms of percentage, with farmers' awareness of AW4 (It is challenging to hire the required number of laborers when necessary), awareness was relatively low (66.7% of farmers), while the other statements (AW5, AW6, and AW7) were relatively high (79.4%, 81.9%, and 80% respectively).
- Farmers were more aware of their time management for planting and harvesting of rice, soil fertility, and proper use of farmyard manure (FYM) and fertilizers.

### 3.2.3. Ministry Management (AW8 through AW10)

- Among the three statements concerned, in terms of percentage, with farmers' awareness of AW8 (Agricultural policies of the Ministry of Agriculture and Irrigation are unstable), awareness was relatively low (56.5% of farmers).
- Farmers had low awareness of agricultural policies. However, they were aware of AW9 (Agricultural extension services are not helpful for farmers) and AW10 (Quality seed is not sufficiently available for farmers).

In summary, maximum awareness (92.1% of farmers) was found in the aspect of general risks, while the lowest awareness (56.5% of farmers) was found in the aspect of Ministry management. In the aspect of Ministry management, there was a remarkable lack of awareness.

### *3.3. Classification of Farmers Based on Their Awareness*

Herein, the findings show the feature of farmers' awareness structure, namely the combination of awareness levels of the ten statements rather than which statements are high or low. According to

the results of cluster analysis by the K-means method, farmers could be classified into three clusters (Table 6), as shown in the dendrogram of Figure 3. The three groups did not differ statistically in terms of many characteristics of farmers, such as age, education, farming experiences, farmland size, active labor force, annual income from crop production, receiving agricultural information, and the number of contacts with extension workers (Table 7). The detailed identification of three clusters is as follows.

**Table 6.** Cluster analysis based on farmers' awareness of the low yield of conventional rice production.

| * Cluster | Mean Value | | | | | | | | | | Number of Farmers (%) |
|---|---|---|---|---|---|---|---|---|---|---|---|
| | AW1 | AW2 | AW3 | AW4 | AW5 | AW6 | AW7 | AW8 | AW9 | AW10 | |
| I | 4.3 | 2.7 | 3.8 | 2.9 | 2.9 | 2.5 | 3.6 | 2.2 | 2.8 | 3.3 | 43 (14%) |
| II | 4.2 | 3.9 | 4.4 | 2.0 | 3.2 | 3.9 | 3.4 | 3.6 | 1.5 | 4.1 | 75 (24%) |
| III | 4.6 | 4.3 | 4.7 | 4.5 | 4.5 | 4.5 | 4.5 | 3.7 | 1.4 | 4.6 | 197 (62%) |

Source: Author. Note: (1) * = Cluster analysis by the K-means method, and (2) AW = awareness.

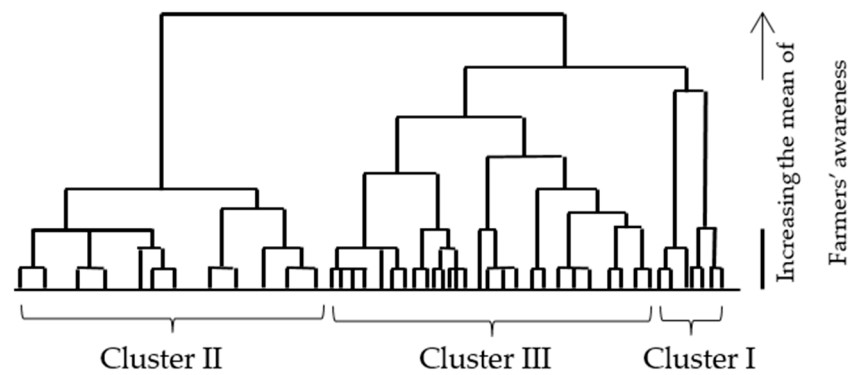

**Figure 3.** Dendrogram of farmers' awareness of low yield of conventional rice production. Source: Author.

**Table 7.** Comparison of farmer characteristics among three clusters.

| Farmers' Characteristics | Cluster I (14% of Farmers) | Cluster II (24% of Farmers) | Cluster III (62% of Farmers) | *p*-Value |
|---|---|---|---|---|
| Age (year) | 47.91 | 49.17 | 51.17 | 0.214 |
| Gender (% of male) | 91 [b] | 96 | 99 [a] | 0.002 |
| Marital status (% of married) | 95 | 89 [b] | 97 [a] | 0.094 |
| Education (year) | 5.07 | 6.01 | 5.51 | 0.304 |
| Farming experiences (year) | 24.14 | 26.25 | 24.27 | 0.511 |
| Household size (person) | 4.23 | 4.08 [b] | 4.74 [a] | 0.005 |
| Access to credit (%) | 88 | 87 [b] | 94 [a] | 0.080 |
| Income from crop production (*kyat/year) | 7,760,802 | 9,993,477 | 7,299,684 | 0.208 |
| Farm size (acre) | 9.37 | 12.48 | 8.70 | 0.112 |
| Active labor force (person) | 3.14 | 3.19 | 3.53 | 0.096 |
| Contact with extension workers (number/year) | 2.12 | 3.33 | 2.86 | 0.220 |
| Receiving agricultural information (%) | 86 | 87 | 89 | 0.817 |
| Membership in local farmers' associations (%) | 35 [b] | 35 [b] | 52 [a] | 0.016 |

Source: Author. Note: (1) * 1 kyat = 0.00067 USD (1 USD = 1492 kyats, as of 9 December 2019). (2) Mean values of clusters denoted by different letters (a, b) show significant differences at the 5% significant level.

### 3.3.1. Cluster I (43 Farmers: 14%)

Since a Likert scale score of four and over means "agree: is aware", their awareness was limited to AW1 (Climate change). Their awareness extended to AW3 (Knowledge of rice production technology is inadequate) and AW7 (Farmers do not use the adequate and correct amount of FYM and fertilizers) only. Therefore, their awareness was narrow but not necessarily high. The farmers in this group shared some characteristics. Their average income was relatively higher than that of Cluster II. Furthermore,

the percentage of male farmers and membership to local farmers' associations were relatively lower than that of Cluster III.

### 3.3.2. Cluster II (75 Farmers: 24%)

Using the Likert scale score, the limitation of their awareness was six statements: AW1, AW2, AW3, AW6, AW8, and AW10, when taking the score >3.5 into account. Meanwhile, it is of interest that their awareness of AW4 (It is challenging to hire the required number of laborers when necessary) and AW9 (Agricultural extension services are not helpful for farmers) were low, compared with the other clusters. Therefore, their awareness comparatively widened, but their unawareness was profound, leading to the problem of a severe gap. The highest average values of farming experiences, farmland size, annual income from crop production, and the number of contacts with extension workers were significant features of farmers in Cluster II.

### 3.3.3. Cluster III (197 Farmers: 62%)

Their awareness was broad and high, meaning that there was no remarkable gap in awareness. Their unawareness was limited to only AW9 (Agricultural extension services are not helpful for farmers). Small landholders and low-income farmers belonged to this cluster, and they had a high awareness of the low yield of conventional rice production. Gender (% of male) and household size were highly significantly different at the 0.01 level by the least significant difference test. Almost only male farmers (99%) were observed in this cluster. The maximum household size was observed in this cluster, and it was different from that of Cluster II. The highest average values of farmers who married and were receiving credit for crop production was found in Cluster III, and these values were different from that of Cluster II. In Cluster III, more than fifty percent of farmers participated in memberships of local farmers' associations, and it was significantly different from that of other clusters.

### 3.3.4. Comparison among Three Clusters

Determinants of clustering were likely to be AW2, AW4, AW5, AW6, AW8, AW9, and AW10. It is of interest that AW9 played the role of demarcating Cluster I from Clusters II and III, though AW 9 (Agricultural extension services are not helpful for farmers) showed a lack of awareness in all clusters.

### *3.4. Comparison of Farmers' Awareness among Three Townships*

Farmers' awareness of low yield of conventional rice production was compared among three townships by analysis of variance (Table 8) and Schiff's test (Table 9). According to the *F* value, AW4, AW5, AW6, and AW7 in farmer management and AW9 in Ministry management were significantly different among the three townships.

**Table 8.** Comparison of awareness statements among three townships.

| Aspect | Statement | Myaungmya (*n* = 105) | | | Einme (*n* = 105) | | | Warkhema (*n* = 105) | | | F-Value |
|---|---|---|---|---|---|---|---|---|---|---|---|
| | | Average | Std. Dev. | Variance | Average | Std. Dev. | Variance | Average | Std. Dev. | Variance | |
| General risks | AW1 | 4.45 | 0.84 | 0.71 | 4.51 | 0.87 | 0.75 | 4.49 | 0.86 | 0.73 | 1.76 |
| | AW2 | 3.93 | 1.07 | 1.14 | 3.95 | 1.16 | 1.35 | 4.15 | 0.97 | 0.94 | 1.35 |
| | AW3 | 4.29 | 0.96 | 0.92 | 4.60 | 0.66 | 0.43 | 4.61 | 0.79 | 0.62 | 0.14 |
| Farmer management | AW4 | 3.40 | 1.43 | 2.03 | 3.76 | 1.36 | 1.84 | 3.78 | 1.31 | 1.71 | 2.60 * |
| | AW5 | 3.91 | 1.22 | 1.48 | 4.24 | 1.07 | 1.14 | 3.82 | 1.38 | 1.90 | 3.36 ** |
| | AW6 | 4.09 | 1.24 | 1.54 | 4.11 | 1.20 | 1.43 | 4.17 | 1.13 | 1.28 | 5.41 *** |
| | AW7 | 3.87 | 1.30 | 1.69 | 4.29 | 0.99 | 0.98 | 4.12 | 1.03 | 1.07 | 3.76 ** |
| Ministry management | AW8 | 3.30 | 1.39 | 1.92 | 3.51 | 1.32 | 1.73 | 3.63 | 1.21 | 1.47 | 0.16 |
| | AW9 | 4.10 | 1.11 | 1.22 | 4.46 | 0.87 | 0.75 | 4.57 | 0.72 | 0.52 | 7.52 *** |
| | AW10 | 4.29 | 0.88 | 0.78 | 4.24 | 1.02 | 1.05 | 4.35 | 1.01 | 1.02 | 0.36 |

Source: Author. Note: (1) AW= awareness, (2) *** = significant at 1% level, (3) ** = significant at 5% level, and (4) * = significant at 10% level.

**Table 9.** Comparison of awareness scores among three townships by Schiff's test.

| Township | AW4 | | AW5 | | AW6 | | AW7 | | AW9 | |
|---|---|---|---|---|---|---|---|---|---|---|
| | T1 | T2 | T1 | T2 | T1 | T2 | T1 | T2 | T1 | T2 |
| T2 | 0.362 * | | 0.324 | | 0.314 ** | | 0.419 ** | | −0.333 ** | |
| T3 | 0.381 * | 0.190 | −0.095 | −0.419 ** | 0.324 ** | 0.095 | 0.257 | −0.162 | −0.457 *** | −0.124 |

Source: Author. Note: (1) AW= awareness, (2) *** = significant at 1% level, (3) ** = significant at 5% level, and (4) * = significant at 10% level. (5) Township; T1 = Myaungmya, T2 = Einme, and T3 = Warkhema.

### 3.4.1. AW4 (It is challenging to hire the required number of laborers when necessary)

Myaungmya township had significantly lower awareness (average Likert scale = 3.40) of AW4, compared with the other two townships. Though the average Likert scale score was less than 4.0 in each township, there was no significant difference between the other two townships.

### 3.4.2. AW5 (Farmers cannot plant and harvest rice at the right time)

Regarding AW5, a significant difference was found only between Einme township and Warkema township. Though the average Likert scale score (3.82) of Warkema township was lower than those in the other two townships, Myaungmya township did not show a significant difference from the other two townships.

### 3.4.3. AW6 (Soil fertility is becoming more inadequate for cropping)

Myaungmya township had significantly lower awareness (average Likert scale score = 4.09) of AW6 compared with the other two townships. Though the average Likert scale score was more than 4.0 in each township, there was no significant difference between the other two townships.

### 3.4.4. AW7 (Farmers do not use the adequate and correct amount of FYM and fertilizers)

A significant difference of AW7 was found only between Myaungmya township (average Likert scale score = 3.87) and Einme township (average Likert scale rating = 4.29). There was no significant difference between Einme township and Warkema township, though the average Likert scale score was more than 4.0 in these two townships.

### 3.4.5. AW9 (Agricultural extension services are not helpful for farmers)

Myaungmya township had significantly lower awareness (average Likert scale score = 4.10) on AW9 compared with the other two townships. Though the average Likert scale score was more than 4.0 in each township, there was no significant difference between the other two townships.

Thus, the awareness of AW4, AW6, and AW9 in Myaungmya township was significantly different from the other two townships. In the awareness of AW7, Myaungmya township was different from only Einme township and was a minor difference. Finally, a minor difference was found in the awareness of AW5: Einme township was more aware than Warkhema township.

The differences in farmers' awareness among three townships were due, to some extent, to the differences in their characteristics (Table 10). Among three townships, the highest average values of farmland size, annual income from crop production, and the number of contacts with extension workers per year were observed in Myaungmya township. These values were significantly different and higher than that of the other two townships: Einme and Warkhema. Therefore, farmers' awareness of three statements—AW4, AW6, and AW9—in Myaungmya were lower than that of the other two townships. Because of the maximum average values of education, access to credit, and household size in Einme township, farmers' awareness on AW7 was different from Myaungmya township and that of AW5 from Warkhema township.

**Table 10.** Comparison of farmers' characteristics among three townships.

| Farmers' Characteristics | Myaungmya | Einme | Warkhema | *p*-Value |
|---|---|---|---|---|
| Age (year) | 48.68 | 50.96 | 51.11 | 0.292 |
| Gender (% of male) | 97.1 | 98.1 | 97.1 | 0.881 |
| Marital status (% of married) | 95.2 | 93.3 | 97.1 | 0.434 |
| Education (year) | 5.19 [b] | 6.23 [a] | 5.29 [b] | 0.043 |
| Farming experiences (year) | 23.37 | 27.29 | 26.03 | 0.117 |
| Household size (person) | 4.63 [a] | 4.77 [a] | 4.13 [b] | 0.010 |
| Access to credit (%) | 86.7 [b] | 99.0 [a] | 89.5 [b] | 0.003 |
| Income from crop production (* kyats/year) | 11,558,195 [a] | 7,615,098 [b] | 4,838,736 [b] | 0.000 |
| Farmland size (acre) | 14.21 [a] | 9.04 [b] | 5.81 [b] | 0.000 |
| Active labor force (person) | 3.30 | 3.47 | 3.42 | 0.669 |
| Contact with extension workers (number) | 4.086 [a] | 2.40 [b] | 2.133 [b] | 0.000 |
| Receiving agricultural information (%) | 87.6 | 85.7 | 90.5 | 0.569 |
| Membership of local farmers' association (%) | 40.0 | 50.5 | 47.6 | 0.292 |

Source: Author. Note: (1) * 1 kyat = 0.00067 USD (1 USD = 1492 kyats, as of 9 December 2019). (2) Mean values of clusters denoted by different letters (a, b) show significant differences at 5% significant level.

### 3.5. Determinants of Farmers' Awareness of the Low Yield of Conventional Rice Production

Among ten statements to measure awareness, in terms of "percentage of farmers", the significant difference between "aware" and "not aware", with the former being less than 80%, was found in AW2, AW4, AW5, and AW8. Thus, the study used four statements of awareness, respectively, as the dependent variable in the binary logit model. The result of the binary logit analysis is presented in Table 11. Since the highest value of variance inflation factors (VIFs) for independent variables was 2.67, there was no multicollinearity among these variables.

Among personal characteristics, age and gender showed a positive correlation with the awareness of AW2 (Less attention is paid to rice production due to the small profit) and AW8 (Agricultural policies of the Ministry of Agriculture and Irrigation are unstable), which is in line with the finding of [30] who found that the age of farmers and gender demonstrate a positive correlation with awareness. Additionally, gender positively predicted the awareness of AW4 (It is challenging to hire the required number of laborers when necessary) and AW5 (Farmers cannot plant and harvest rice at the right time), which is in line with the finding of [22]. This is because most of the household heads were male, and they were more actively involved in farming than females. However, farming experience was negatively associated with AW8 (Agricultural policies of the Ministry of Agriculture and Irrigation are unstable). It appears that farmers who have higher farming experience are less aware of agricultural policy.

In economic characteristics, the determinant for both AW2 (Less attention is paid to rice production due to the small profit) and AW8 (Agricultural policies of the Ministry of Agriculture and Irrigation are unstable) was income from crop production. Income from crop production showed a negative correlation with AW2 and AW8. Since farmers have a larger farmland size, they have received relatively higher annual income from crop production. Therefore, these farmers are also less aware of low profit from rice production. Farmers, who have higher income from crop production are less aware of agricultural policies. They did not note that agricultural policies of the Ministry of Agriculture and Irrigation were unstable.

Among farming characteristics, the determinant of awareness of AW4 (It is challenging to hire the required number of laborers when necessary) was farmland size. This is in line with the findings of [30]. The result explains that farmers who have larger farmland size have a lower awareness of the problems of the hired labor force. This is likely because farmers who have larger farmland size can pay the wages before the rice-growing season so that they can easily hire a labor force when they need rice production. On the other hand, household size had a positive correlation with the awareness of AW4. This is likely because more members of the household were engaged in rice production, enabling them to operate farming practices efficiently.

**Table 11.** The estimated coefficients of the binary logit model.

| Independent Variables | AW2 | | AW4 | | AW5 | | AW8 | |
|---|---|---|---|---|---|---|---|---|
| | Coef. | SE | Coef. | SE | Coef. | SE | Coef. | SE |
| Constant | −2.266 | 1.345 | −2.570 | 1.196 | −2.9512 | 1.337 | −2.932 | 1.390 |
| Age ($X_1$) | 0.069 *** | 0.023 | 0.025 | 0.017 | 0.0312 | 0.022 | 0.047 *** | 0.167 |
| Gender ($X_2$) | 2.687 *** | 0.920 | 1.338 * | 0.809 | 1.821 ** | 0.834 | 2.403 ** | 1.135 |
| Marital Status ($X_3$) | −0.475 | 0.701 | 0.311 | 0.608 | 0.278 | 0.718 | −0.413 | 0.631 |
| Education ($X_4$) | 0.021 | 0.049 | −0.002 | 0.042 | 0.036 | 0.050 | −0.001 | 0.040 |
| Farming experience ($X_5$) | −0.023 | 0.020 | −0.016 | 0.015 | −0.028 | 0.019 | −0.030 ** | 0.015 |
| Household size ($X_6$) | 0.230 | 0.151 | 0.273 ** | 0.127 | 0.266 * | 0.153 | 0.172 | 0.118 |
| Access to credit ($X_7$) | 0.511 | 0.508 | 0.160 | 0.463 | 0.172 | 0.492 | −0.137 | 0.450 |
| Income from crop production ($X_8$) | −0.369 * | 0.208 | 0.087 | 0.193 | −0.213 | 0.233 | −0.441 ** | 0.180 |
| Farmland size ($X_9$) | 0.015 | 0.014 | −0.024 * | 0.014 | −0.015 | 0.013 | 0.013 | 0.012 |
| Active labor force ($X_{10}$) | −0.185 | 0.173 | −0.107 | 0.148 | 0.094 | 0.186 | −0.117 | 0.139 |
| Contact with extension workers ($X_{11}$) | −0.057 | 0.039 | −0.042 | 0.043 | 0.031 | 0.046 | 0.035 | 0.039 |
| Receiving agricultural information ($X_{12}$) | −1.680 ** | 0.733 | −0.451 | 0.431 | −0.093 | 0.487 | −0.331 | 0.392 |
| Membership of local farmers' association ($X_{13}$) | −0.211 | 0.298 | 0.424 | 0.263 | 0.458 | 0.312 | 0.164 | 0.245 |
| Location; Einme township ($X_{14}$) | −0.380 | 0.368 | 0.343 | 0.326 | 0.605 | 0.426 | 0.314 | 0.311 |
| Location; Warkhema township ($X_{15}$) | 0.093 | 0.402 | 0.381 | 0.339 | −0.552 | 0.382 | 0.134 | 0.323 |

Source: Author. Note: AW = awareness, Coef. = coefficient, SE = standard error, *** = significant at 1% level, ** = significant at 5% level, and * = significant at 10% level.

Among institutional characteristics, the determinant of AW2 (Less attention is paid to rice production due to the small profit) was negatively correlated with receiving agricultural information. Even though farmers have received agricultural information that was delivered by the Department of Agriculture and agrochemical companies, they were less aware of low profit from rice production. Perhaps this is because they have received basic knowledge and did consider the benefit. It is of interest that location was not a significant determinant of awareness of the low yield of conventional rice production.

## 4. Conclusions

Among the three aspects of awareness, most of the farmers were aware of the aspect of general risks. However, some farmers have low awareness of the aspect of farmer management and Ministry management. In the aspect of farmer management, only two-thirds of farmers were aware of challenging labor problems. In the Ministry management aspect, around half of farmers were not aware of agricultural policies. Farmers were categorized into three clusters based on their awareness. Among three clusters, more than half of the farmers were in Cluster III and showed a broader structure of awareness. Farmers in Cluster III had large household sizes, received credit for crop production, and had membership in local farmers' associations. Among three townships, farmers who lived in Myaungmya township had a lower awareness of challenging labor problems, poor soil fertility, and unhelpful extension services compared that of other townships. The finding is somewhat unintuitive, as their farming (larger farmland size), economic (higher income), and institutional characteristics (contact with extension workers) contributed to decreased awareness.

As a whole, personal characteristics (particularly gender, age, and household size) determine farmers' awareness of the low yield of conventional rice production. The study found that farmers' awareness was negatively associated with farming experiences, where higher income levels of farmers, a larger farmland size, and receiving agricultural information were associated with low awareness. Farmers who had more farming experience were satisfied with the return of rice from conventional production. Some farmers received higher total income from crop production because they had a larger farmland size, and they were less aware of the low yield of conventional rice production. According to farmers, even though they received agricultural information, some information was not necessarily related to rice production. The finding implies that there are weaknesses in the current extension service programs on the dissemination of the GAP to farmers. Farmers' awareness of the low yield of conventional rice production can be increased through developing extension service programs to distribute useful information on rice production effectively.

**Funding:** This research was financially supported by YAU-JICA Technical Cooperation Project, Myanmar.

**Acknowledgments:** The author is very grateful to the Japan International Cooperation Agency (JICA) for offering the scholarship for doctoral study at the Graduate School of International Development (GSID), Nagoya University, Japan. The author would like to show his gratitude to YAU-JICA Technical Cooperation Project for supporting all expenditures of this research.

**Conflicts of Interest:** The author declares no conflict of interest.

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
