# Peer review of "Farmers’ Awareness of the Low Yield of Conventional Rice Production in Ayeyarwady Region, Myanmar: A Case Study of Myaungmya District"

_agriculture, doi:10.3390/agriculture10010026_

Round 1

Reviewer 1 Report

The found the analytical procedures and analyses of data to be scientifically sound and appropriate for the purpose of the study. The presentation of results were also complete and well presented.

However, I found the basic design of the study, and therefore the conclusions, to be somewhat lacking in terms of insights into potential problems that might undermine the conclusions.

Specifically, the basic question is in this study is farmers' awareness, "defined as a state of knowing the reasons for the low yield of conventional rice production." The questions chosen for the survey were "based on a pilot survey and key informants interviews on the causality of low yield." The validity of the conclusions depends entirely on whether the actual reasons for low yield in conventional rice production are accurately represented by the reasons identified in the survey, for which no further justification was provided. The general conclusion was that farmers had inadequate awareness of causes of low rice yields on conventional farms. An alternative conclusion is that farmers are aware that actual causes of low yields on their farms, or the actual potentials for higher yields, and that actual causes are different from those identified in the survey questions. Their answers may be more a reflection of "disagreement" than of "lack of awareness." The specific groups identified as having "less awareness" would tend to support this alternative hypothesis. "Higher farming experiences, higher income levels of farmers, larger farmland size, and receiving agricultural information were associated with low awareness."

In my opinion, this finding provides just cause to question the basic design of the experiment, at least the questions used to identify farmers' awareness, or at least to seriously question the conclusions.    

Author Response

Dear Reviewer 1,

Thank you very much for your comments and suggestions. I already responded to your comments as an attachment.

Thank you very much for your kind attention.

Sincerely,

Soe Paing Oo

Reviewer 2 Report

1. Abstract 

The abstract should briefly introduce the problem statement and the methodologies used.  

2. Introduction 

In page 2 the author states: "There are five steps of adoption: awareness, interest, trial, evaluation, and adoption[8]. Among them, awareness is an essential factor for the dissemination of environmental knowledge and communication of its fundamental elements[9]. It is necessary to understand the technology adoption process of farmers so that the GAP of rice can disseminate in Myanmar. There is a need to study farmers’ awareness before attempting the adoption of new technology[8]. As the first step of the process, farmers must be aware of problems that relate to farming practices. If farmers notice their problems, they will search for the proper solution to the problems." But this is not enough to conduct the study. Hence, I recommend the author to provide the existing kinds of literature gap in relation to awareness issues from crop technology adoption standpoint.   

3. Methodology

This section is very clear! But the reason why clustering is needed is sufficiently addressed 

4. Results and Discussion 

Under each table (Source: Field Survey Data.) indicate the year when the file survey was conducted. Write the full Statement of each awareness rated item in Table 5(Table 5. Comparison of mean values of farmers who not aware and aware in each statement of awareness) 

It is not clear why section 3.3. The classification of farmers based on their awareness is needed.

5. Conclusion 

This section is weak. It needed reworking by avoiding duplication of sentences written in the discussion sections 

Author Response

Dear Reviewer 2,

Thank you very much for giving valuable comments and suggestions. Based on your comments, I already responded to your comments as an attachment.

Thank you for your kind attention.

Sincerely,

Soe Paing Oo

Round 2

Reviewer 1 Report

My primary concern was with the conclusions. I question whether conclusions regarding participants lack of knowledge might reflect a lack of agreement with statements rather than lack of knowledge of the stated conclusions.  Apparently, the authors have considered my question and have reaffirmed that their original conclusions are correct.